# Inhibitory Effects of Varespladib, CP471474, and Their Potential Synergistic Activity on *Bothrops asper* and *Crotalus durissus cumanensis* Venoms

**DOI:** 10.3390/molecules27238588

**Published:** 2022-12-06

**Authors:** Sara Quiroz, Isabel C. Henao Castañeda, Johan Granados, Arley Camilo Patiño, Lina María Preciado, Jaime Andrés Pereañez

**Affiliations:** 1Research Group in Toxinology, Pharmaceutical, and Food Alternatives, Pharmaceutical and Food Sciences Faculty, University of Antioquia, Medellín 50010, Colombia; 2Research Group in Marine Natural Products, Pharmaceutical and Food Sciences Faculty, University of Antioquia, Medellín 050010, Colombia; 3Research Group in Pharmaceutical Promotion and Prevention, Universidad de Antioquia, Medellín 050010, Colombia

**Keywords:** *Bothrops asper*, Colombian rattlesnake, snake venom, Phospholipase A_2_, snake venom metalloproteinase, inhibitors, molecular docking, synergistic antivenoms

## Abstract

Snakebite is a neglected tropical disease that causes extensive mortality and morbidity in rural communities. Antivenim sera are the currently approved therapy for snake bites; however, they have some therapeutic limitations that have been extensively documented. Recently, small molecule toxin inhibitors have received significant attention as potential alternatives or co-adjuvant to immunoglobulin-based snakebite therapies. Thus, in this study, we evaluated the inhibitory effects of the phospholipase A_2_ inhibitor varespladib and the metalloproteinase inhibitor CP471474 and their synergistic effects on the lethal, edema-forming, hemorrhagic, and myotoxic activities of *Bothrops asper* and *Crotalus durissus cumanensis* venoms from Colombia. Except for the preincubation assay of the lethal activity with *B. asper* venom, the mixture showed the best inhibitory activity. Nevertheless, the mix did not display statistically significant differences to varespladib and CP471474 used separately in all assays. In preincubation assays, varespladib showed the best inhibitory activity against the lethal effect induced by *B. asper* venom. However, in independent injection assays, the mix of the compounds partially inhibited the lethal activity of both venoms (50%). In addition, in the assays to test the inhibition of edema-forming activity, the mixture exhibited the best inhibitory activity, followed by Varespladib, but without statistically significant differences (*p* > 0.05). The combination also decreased the myotoxic activity of evaluated venoms. In these assays, the mix showed statistical differences regarding CP471474 (*p* < 0.05). The mixture also abolished the hemorrhagic activity of *B. asper* venom in preincubation assays, with no statistical differences to CP471474. Finally, the mixture showed inhibition in studies with independent administration in a time-dependent manner. To propose a mode of action of varespladib and CP471474, molecular docking was performed. PLA_2s_ and SVMPs from tested venoms were used as targets. In all cases, our molecular modeling results suggested that inhibitors may occupy the substrate-binding cleft of the enzymes, which was supported by specific interaction with amino acids from the active site, such as His48 for PLA_2s_ and Glu143 for the metalloproteinase. In addition, varespladib and CP471474 also showed interaction with residues from the hydrophobic channel in PLA_2s_ and substrate binding subsites in the SVMP. Our results suggest a synergistic action of the mixed inhibitors and show the potential of varespladib, CP471474, and their mixture to generate new treatments for snakebite envenoming with application in the field or as antivenom co-adjuvants.

## 1. Introduction

Snakebite envenoming constitutes a relevant public health problem in tropical and subtropical regions of the world [1,2]. It is estimated that about 4.5–5.4 million people are bitten by snakes every year, out of which 1.8–2.7 million develop envenoming, which leads to approximately 81,000–138,000 deaths. In addition, there is no clear information on the number of patients that suffer sequelae that include physical and psychological problems (amputations, contractures, blindness, chronic kidney disease, post-traumatic stress disorder, and malignant ulcers, among others) [1,3]. Therefore, snake envenomation was added to the list of neglected tropical diseases by the WHO in March 2009 and later removed [4]. However, with sufficient epidemiological data, the menace was re-included in category A of neglected tropical diseases in June 2017 [5]. Furthermore, the WHO proposed a plan to reduce snakebite deaths and disabilities by 50% before 2030 [6].

In Colombia, for the year 2021, 4,644 snakebites were reported [7]. The species from the *Bothrops* genus are responsible for the highest number of bites, owing to their wide distribution in Colombia [8,9]. Snakebites inflicted by these species cause local effects, such as myonecrosis, hemorrhage, dermonecrosis, pain, blister formation, and edema. In contrast, systemic manifestations involve hemodynamic disturbances, coagulopathies, systemic hemorrhage, and acute kidney injury (AKI) [8]. The most abundant components of *B. asper* venom are phospholipases A_2_ (PLA_2s_), snake venom metalloproteinases (SVMPs), and serine proteinases [10,11], which are responsible for the pathophysiological disorders previously mentioned. On the other hand, the Colombian rattlesnake *Crotalus durissus cumanensis* causes 1–3% of snakebites in Colombia [8]. The snakebites inflicted by this species are characterized by mild local effects, which include pain and edema. This differs from envenoming induced by other viperids for being predominantly neurotoxic and myotoxic. The systemic alterations comprise neurotoxicity (flaccid paralysis in the respiratory musculature and other peripheral muscles) and myotoxicity that might lead to rhabdomyolysis, AKI, and hemostatic disturbances [8]. The most abundant component in the venom of *C. d. cumanensis* is crotoxin (CTX, 64.7%) [12], which is a heterodimer composed of a basic enzymatically active PLA_2_ (CB), and an acidic non-enzymatic chain (CA or crotapotin). The crotapotin acts as a chaperone by blocking the active site of CB and reducing its catalytic activity but enhancing its specificity for the presynaptic neuromuscular junction, inhibiting the release of acetylcholine, which induces flaccid paralysis [13,14,15]. Moreover, the venom of *C. d. cumanensis* also contains 3.3% of SVMPs [12].

The only available and accepted therapy for treating snakebite envenoming is the intravenous (IV) administration of animal-derived antivenoms, composed of purified Immunoglobulins-G (Ig-G) derived from the plasma of large animals following immunization with snake venom. Nevertheless, several issues limit the efficacy and accessibility of antivenoms. For instance, these products are not available in rural areas, in which most of the accidents occur, they must be administered in healthcare facilities by health professionals, and it has been demonstrated that antivenoms have limited efficacy in neutralizing local tissue damage induced by snake venoms [16,17]. Therefore, there is a need to research new inhibitory molecules that could complement antivenoms by applying them after the onset of the envenoming in the field.

In recent years, small molecule therapeutics have become a focus of research for their ability to inhibit SVMPs and PLA_2s_, which may help complement antibody-based antivenoms [18]. For instance, varespladib has been the most studied compound for inhibiting snake venom PLA_2s_. This molecule was developed as a potent inhibitor of human secreted PLA_2s_ for treating pancreatitis, sepsis, rheumatoid arthritis, and cardiovascular diseases [19,20]. However, varespladib and methyl varespladib (the pro-drug of varespladib active via oral) inhibited the PLA_2_ activity of 28 snake venoms of medical importance [21]. Furthermore, other studies demonstrated that varespladib and Methyl varespladib abolished the lethal activity of *Oxyuranus scutellatus* and *Micrurus fulvius* venoms in mice and pigs [22,23]. In addition, varespladib also inhibited purified PLA_2s_ from different snake venoms [24,25]. More recently, it was reported that varespladib inhibits the coagulotoxic effects of several venoms [26,27,28,29].

On the other hand, the compound CP471474 (Pfizer Global Research and Development) was designed as a broad-spectrum matrix metalloproteinase inhibitor. Its structure displays a pyran-containing sulfonamide hydroxamate [30]. Previously, this molecule was reported by its capacity to inhibit a PI-SVMP from *B. atrox* venom, demonstrating its ability to reduce the enzymatic and hemorrhagic activities induced by this toxin, with an IC_50_ of 11.6 and 2.5 μM, respectively. In addition, the mode of action of this compound was described by molecular dynamic simulations, indicating that CP471474 interacts with the metalloproteinase substrate binding cleft [31].

Recently, a study was reported on snake venom inhibition with a combination of inhibitors for the most abundant toxins in viperid venoms, i.e., PLA_2_ and SVMP_s_. Albulescu et al. [32] reported the inhibition of the lethal activity of *Echis occelatus*, *E. carinatus*, *B. asper* (from Costa Rica), and *Daboia russelli* venoms by a combination of marimastat (an SVMP inhibitor) and varespladib. However, this kind of study has not been reported for snake venoms from Colombia. Therefore, it is important to consider variations in their toxin’s relative abundance and specificity between different species. These variations depend on parameters such as age, geographic location, time of the year, and the envenoming snake’s diet, among others [33]. Therefore, to study the inhibitory potential of a combination of compounds targeting PLA_2_ and SVMP_s_ with snake venoms from Colombia, the present study aimed to evaluate the inhibitory effects of varespladib and CP471474 on the lethal, myotoxic, edema-forming, and hemorrhagic activities of Colombian *B. asper* (mapaná) and *C. d. cumanensis* (rattlesnake) venoms. In addition, a mix of the mentioned compounds was also tested and compared with the inhibitory properties of compounds used separately to evaluate a possible synergy.

## 2. Results

### 2.1. Inhibition of Lethal Activity in Preincubation Assays

Varespladib, CP471474, and their mixture containing 10 mg/kg of mice of each compound, were tested in their ability to inhibit the lethal activity of the Colombian *B. asper* and *C. d. cumanensis* venoms (Figure 1). In the preincubation assays, venoms killed all mice. In the assay with *B. asper* venom, in groups that received venom and varespladib, venom plus a mix of the compounds, four and three mice survived, respectively (*p* < 0.05, with respect to positive control). Whereas in groups that received venom plus CP471474, one mouse was alive at the end of the experiment (*p* > 0.05 regarding venom alone, and the other treatment groups). Hence, varespladib inhibited 100% of the lethal effect of *B. asper* venom (Figure 1A). In contrast, in the assay with *C. d. cumanensis* venom, in the groups that received venom plus a mix of the compounds, three mice survived (*p* < 0.05, with respect to venom alone). In the groups that received venom and varespladib or venom plus CP471474, one or two mice remained alive (*p* > 0.05 with respect to positive control). In addition, there were no statistically significant differences between groups containing one or two inhibitors (*p* > 0.05) (Figure 1B). In both cases, the compound mix inhibited 75% of the lethal activity of the venoms.

### 2.2. Inhibition of Lethal Activity in Independent Injection Assays

In these assays, the inhibitors were immediately injected intraperitoneally after venom injection by the same route. In both tests, compounds and their mix partially inhibited the lethal activity of the *B. asper* and *C. d. cumanensis* venoms (Figure 2). In the assay with *B. asper* venom and either CP471474 or varespladib, it registered one live mouse, respectively (*p* > 0.05 concerning venom alone) (Figure 2A). On the other hand, in the assay with *C. d. cumanensis* venom, varespladib showed two live mice and CP471474 one live mouse at the end of the assay, but there were no statistical differences to venom (*p* > 0.05) (Figure 2B). In both assays, the mix of the compounds inhibited 50% of the lethal activity of the venoms. Nonetheless, these results were not significantly different from venoms and separate compounds (*p* > 0.05) (Figure 2A,B).

### 2.3. Inhibition of Edema-Forming Activity in Independent Injection Assays

The capacity of varespladib, CP471474, and their mix to inhibit the edema-forming activity of the *B. asper* and *C. d. cumanensis* venoms were tested in independent injection assays (Figure 3). In the assays with both venoms, the mix of the compounds showed the best inhibition ability (*p* < 0.05 with respect to venom alone), followed by varespladib (*p* < 0.05 with respect to venom alone) (Figure 3A,B). Nevertheless, there were no statistically significant differences between these two treatments. In addition, in both assays, CP471474 did not show statistically significant differences with respect to venoms (*p* > 0.05) (Figure 3A,B).

### 2.4. Inhibition of Myotoxic Activity in Preincubation Assays

The inhibition of the myotoxic activity of Colombian *B. asper* and *C. d. cumanensis* venom, by varespladib, CP471474, and their mixture was tested in preincubation assays (Figure 4A,B). The mixture of the compounds displayed the best capacity to reduce the plasma CK activity induced by both venoms (*p* < 0.05 with respect to venoms). In the assays, the mix of the compounds showed statistically significant differences regarding CP471474 (*p* > 0.05) (Figure 4A,B).

### 2.5. Inhibition of the Hemorrhagic Activity of B. asper venom in Preincubation and Independent Injection Assays

In the preincubation assays, the mix of the compounds and CP471474 abolished the hemorrhagic activity induced by *B. asper* venom (*p* < 0.05 concerning venom alone and *p* > 0.05 between them) (Figure 5A). In this assay, varespladib partially inhibited the hemorrhage induced by *B. asper* venom. Furthermore, when varespladib, CP471474, or a mix of them were injected immediately after venom injection (time = 0 min), all of them partially decreased the hemorrhagic diameter induced by the venom (*p* < 0.05 respect to venom alone). Nevertheless, there were no statistically significant differences between all treatments (*p* > 0.05) (Figure 5B).

### 2.6. Inhibition of the Hemorrhagic Activity of B. asper venom with Independent Injection at Different Time Intervals

In this assay, the compound mix was injected at different intervals (0, 3, and 5 min) after venom injection. The combination partially abrogated the hemorrhagic activity of the venom (Figure 6). In addition, this inhibition was time dependent. At all times, the compound mix displayed statistical differences with respect to the positive control (*p* > 0.05). However, at time zero minutes, the combination showed statistically significant differences with respect to the other times (*p* < 0.05).

### 2.7. Molecular Docking Studies

With the aim of suggesting the inhibition mechanism of the SVMP and PLA_2_ toxins, docking studies were performed with the evaluated compounds. The crystal structures of the metalloproteinase BaP1 (PDB code 2W15), the PLA_2_ myotoxin I (MT-I) (PDB code 5TFV_A), both isolated from *B. asper* venom, and the PLA_2_ from the crotoxin complex (PDB code 2QOG_B), isolated from *C. durissus terrificus* venom, were selected as receptors. Docking conformations with the lowest affinities were chosen and described. Affinities and the main protein–ligand interactions are shown in Table 1. In addition, the location of the compounds in the protein pockets is displayed in Figure 7 and Figure 8.

Our docking results suggested that varespladib exhibited the highest affinity with the PLA_2_ from *C. d. terrificus* and the lowest with myotoxin I from *B. asper*. Moreover, varespladib might interact with the active site of the enzymes and occupy their hydrophobic channel (Figure 7B,D and Figure 8B,D). In addition, varespladib also displayed interaction with the active site of the SVMP BaP1 from *B. asper* venom and the sub-sites of the substrate binding cleft (Figure 7F and Figure 8F).

On the other hand, the computational simulation showed the best affinity between CP471474 and the metalloproteinase BaP1 from *B. asper* venom. This interaction is supported by interactions with amino acids belonging to the zinc-binding motif (His142) and residues from the substrate binding subsites (Figure 7E and Figure 8E). Additionally, CP471474 also might interact with PLA_2s_ with lower affinities. Still, it might bind to Asp49, a residue that coordinates the Ca^2+^ cofactor, and interact with residues from substrate binding clef of these enzymes (Figure 7A,C and Figure 8A,C).

## 3. Discussion

Snakebite is one of the most lethal tropical neglected diseases, with about 138,000 deaths yearly. It mainly affects poor people in the rural areas of tropical and subtropical regions of the world, where antivenoms are unavailable [1,2]. In addition, this product also has other problems, such as mandatory administration in clinical facilities, poor cross-species neutralization, and limited efficacy in neutralizing local tissue damage induced by snake venoms, among others [16,17]. Therefore, new strategies to improve therapy access, such as venom characterization, antivenomic studies [34], humanized antibodies [35], and searching for small molecule inhibitors for snake venoms [21,22,23,24,25,26,27,28,29,31,32,36], have been performed.

SVMPs and PLA_2s_ are the most abundant and important toxins in snake venoms from the Viperidae family [1,10,11,12,37]. Consequently, we chose two small molecules with inhibitory activity on these main toxins. Varespladib has been previously tested against several snake venoms and toxins [21,22,23,24,25]. In this study, varespladib partially inhibited the lethal activity of *C. d. cumanensis* venom (50%) in either preincubation or independent injection assays. The neurotoxicity caused by this venom is due to the presence of crotoxin, a heterodimeric PLA_2_. This toxin is also present in the venom of the South American *C. durissus* sub-species. In previous studies, varespladib delayed the lethal activity of *C. d. terrificus* venom until six hours [38]. However, after 24 h, the inhibitor did not protect the mice, and all the animals died. Nevertheless, these assays were performed in a rescue model, i.e., subcutaneous venom injection and intravenous or oral administration of varespladib. Therefore, our results are not comparable to Gutierrez et al. [38]. The inhibitory activity of varespladib on the *C. durissus* sub-species venoms, including *C. d. cumanensis* may be explained by the ability of this compound to inhibit the crotoxin complex, its PLA_2_, in their capacity to induce neuromuscular blockade and myotoxicity [39], which agrees with the inhibition of myotoxicity and edema-forming activity induced by *C. d. cumanensis* reported in our study.

In contrast, varespladib inhibited the lethal effect of *B. asper* venom in preincubation assays, but it only protected 25% of mice when it was injected independently with venom. Nevertheless, the inhibitory effect of varespladib on the lethal activity of *B. asper* venom has not been reported previously. Moreover, the partial inhibition capacity of this molecule on the edema-forming activity induced by *B. asper* venom was also reported in this study. The inhibition of this compound on *B. asper* whole venom may be explained by its neutralizing capacity on enzymatic and myotoxic activities of myotoxin I (a PLA_2_) from *B. asper* venom [24]. In addition, it has been reported that varespladib also inhibited the coagulotoxic effects of PLA_2s_ present in the *B. asper* venom and partially abrogated the procoagulant activity induced by other toxins [29], including SVMPs. The inhibition of venom metalloproteinases by varespladib is an important finding; it can explain the partial inhibition of hemorrhage induced by *B. asper* venom reported herein. These results agree with those reported by Gutierres et al. [40] and Wang et al. [41], who informed the inhibitory activity of this molecule on the hemorrhage induced by *Lachesis muta rhombeata*, and *Deinagkistrodon acutus* and *Agkistrodon halys* venoms, respectively.

On the other hand, CP471474 is a matrix metalloproteinase inhibitor with a pyran-containing sulfonamide hydroxamate structure [30]. This molecule was initially tested against snake venoms and toxins in our lab. We reported that CP471474 inhibited proteolytic, hemorrhagic, and edema-forming activities of *B. atrox* venom and its main SVMP [31]. These previous results agree with the inhibition of the hemorrhagic activity induced by *B. asper* venom. Moreover, CP471474 partially inhibited (25%) the lethal activity of *B. asper* and *C. d. cumanensis* venoms, in either preincubation or independent injection assays. This is the first report about the inhibition of CP471474 on lethal activities of snake venoms. Nevertheless, CP471474 lacks inhibitory activity on myotoxicity induced by *B. asper* venom.

To propose a possible mode of action for the inhibitors tested in this study, we performed molecular docking studies of compounds against myotoxin I (a PLA_2_) and BaP1 (an SVMP) from *B. asper* venom; and CB (a PLA_2_) from crotoxin complex from *C. d. terrificus* venom. As expected, our results suggested favorable interactions of varespladib with PLA_2s_. This compound exhibited interactions with His48, an amino acid responsible for the water deprotonation that generates the hydroxyl moiety that will attack sn2 position of the glycerophospholipid [42,43]. Thus, the described interaction may block the catalytic cycle of the PLA_2s_. In addition, varespladib also showed interaction with Asp49, which is mainly involved in cofactor (Ca^2+^) coordination and transition-state stabilization [42,43]. Moreover, the inhibitor also displayed weak interactions with residues of the interfacial binding surface and hydrophobic channel (Leu2, Phe5, Ala18, Ile19, Trp31, Phe24, Lys69 in CB from crotoxin complex; and Leu2, Phe5, Ala6, Ile9, Phe19, from myotoxin I) [42,43,44]. To summarize, the interactions described above may explain the inhibition of varespladib on the catalytic cycle and the substrate recognition of snake venom PLA_2s_.

Conversely, docking results for CP471474 showed the highest affinity with BaP1 and lower affinities with the PLA_2s_ structures. This result was expected since CP471474 has been reported as a metalloproteinase inhibitor [31], and experimental results in this study showed this compound is able to abolish the hemorrhagic activity of *B. asper* venom. Our docking results suggested that CP471474 showed interactions with the amino acids Thr107 and Arg110 (hydrogen bonds), which belong to S1 substrate-binding subsite, Ser168, Val169, and Leu170, which are in S1’ subsite, and Ile108 from S3 subsite [31,45]. Consequently, all the interactions described above may affect the substrate binding for the enzyme. In addition, interactions with His142, a structural amino acid that helps to bind Zn^2+^ ion, may disturb the cofactor coordination [31,45].

As mentioned above, a relevant result obtained in this study was the inhibitory capacity of varespladib on the hemorrhagic activity induced by *B. asper* venom. Molecular modeling results suggested that varespladib may bind Thr107 through a hydrogen bond (S1 subsite) [31,45]. In addition, the inhibitor may also interact with Ser168, Val169, Leu170 (s1’subsite), and Ile108 (S3 subsite). These interactions may explain the BaP1-substrate-binding cleft occupation by varespladib [31,45]. In addition, varespladib also may dislocate the cofactor coordination, which is supported by the interaction observed with His142 [31,45]. Another critical fact contributing to the SVMP inhibition of varespladib is the van der Waals connection with Glu143. This residue is essential to the metalloproteinase catalytic mechanism since the carboxylate group of Glu143 facilitates the peptide bond hydrolysis in the active site by deprotonating the catalytic water and generating a hydroxyl moiety that acts as a nucleophile to attack the carbonyl carbon of the peptide bond on the substrate [31,45]. All the interactions described above may explain the inhibition of the catalytic cycle of SVMPs present in *B. asper* venom and the occupation of the substrate-binding cleft by varespladib.

To have insights into venom inhibition by a mix of PLA_2s_ and SVMPs inhibitors, we tested the inhibitory capacity of a mixture containing 10 mg/kg of each compound against lethal, edema-forming, myotoxic and hemorrhagic activities of Colombian snake venoms. Nevertheless, this study has no comparable data since this is the first report with the compound CP471474 used in a mix. The mixture of these compounds partially inhibited the lethal activity of the *B. asper* venom and *C. d cumanensis* venoms in either preincubation (75%) or independent injection (50%) assays. However, the mix did not show statistical differences (*p* > 0.05) regarding the inhibition obtained with varespladib and CP471474 used separately. Survival of mice after envenomation with an injection of a mix of varespladib and an SVMPs inhibitor (marimastat) has also been reported by Albulescu et al. [32]. The mixture used in our study also inhibited the local tissue damage activities of the tested venoms. For instance, the combination inhibited the edema-forming activity of *B. asper* and *C. d. cumanensis* venoms. These assays showed significant statistical differences from when CP471474 was used alone. The edema-forming activity induced by snake venoms requires an interplay of several toxins [46,47,48]. However, it appears that in *B. asper* and *C. d. cumanensis* venoms from Colombia, the significant contribution is due to the PLA_2_ activity. This observation is supported since varespladib alone showed comparable inhibition activity with the mix. Another important result of the mixture was the partial inhibition of myotoxic activity induced by *B. asper* and *C. d. cumanensis* venoms (*p* < 0.05 with respect to venoms). In addition, the combination abolished the hemorrhage caused by *B. asper* venom in the preincubation assays. This result was like that obtained with CP471474 used alone. Moreover, the mixture also inhibited the hemorrhage in a time-dependent manner. In this effect, as mentioned above, a particular contribution of varespladib should be considered.

## 4. Materials and Methods

### 4.1. Venoms

A venom pool was obtained by manual extraction from *B. asper* and *C d cumanensis* adult specimens from the subregion of Antioquia Magdalena Medio and Meta, the southeast region of Colombia, respectively. Snakes were maintained in captivity at the Serpentarium of the Universidad de Antioquia (Medellín, Colombia). Venoms were centrifuged at 800× *g* for 15 min, and supernatants were lyophilized and stored at −20 °C until used.

### 4.2. Chemicals and Reagents

The compounds CP471474 and varespladib were purchased from Sigma-Aldrich. CP471474 was dissolved in saline solution, and varespladib was dissolved in DMSO (10 mg/mL) due to solubility issues.

### 4.3. Animals

Male and female Swiss Webster mice (18–20 g) were used for in vivo experiments. All animals received food and water ad libitum under controlled environmental conditions. The experimental protocol was approved by the institutional Committee for the Use and Care of Research Animals at the Universidad de Antioquia (Protocol code 139 from 21 March 2021).

### 4.4. Inhibition of the Lethal Activity

For the lethality inhibition experiments, 3 DL_50_ of *B. asper* (186 µg) and *C. d. cumanensis* (5.4 µg) venoms were used as a positive control. The treatments were the individual compounds CP471474 and varespladib, in a dose of 10 mg/kg, preincubated with each venom for 30 min at 37 °C. For the mixture, compounds also were used in a dose of 10 mg/kg. As a negative control, the mixed compounds (Negative mix control) and saline solution (negative control) were injected. The injection volume was 300 µL via intraperitoneal for all the experiments, and four mice were used for each treatment and control. Death and surviving mice were counted past 24 h. The survivor mice were euthanized by carbon dioxide inhalation. For independent injection experiments, groups of four mice were injected intraperitoneally with 3 DL_50_ of each venom, and immediately later, the mixed inhibitors were injected using a dose of 10 mg/kg [48].

### 4.5. Inhibition of Edema-Forming Activity

*C.d cumanensis* and *B. asper venoms* (5 μg in 50 μL saline solution) were administered subcutaneously in the left footpad of groups of four mice. The right footpad, injected with 50 μL of saline solution, was used as a negative control. Later, each group of four mice was injected with the inhibitor CP471474 or varespladib, in a dose of 10 mg/kg, intramuscularly into the right gastrocnemius muscle. Another group received a mix of the compounds at the same amount. Edema was assessed three hours after injection by measuring the footpad thickness with a caliper and expressed as the difference between the thickness of the left and right footpad [49].

### 4.6. Inhibition of Myotoxicity

For myotoxicity studies, venom (100 uL) was injected by the intramuscular route into the gastrocnemius of three mice. *B. asper* control group received 50 µg of venom. For *C. d. cumanensis*, the dose was 10 µg. These doses were selected, considering the activity of the assessed venoms.

For the treatments, two groups received the compounds CP471474 or varespladib at a dose of 10 mg/kg using the same route for venom. A third group received a mix of the inhibitors (10 mg/kg). Negative controls were injected with a mix of inhibitors or saline solution. All experimental doses were preincubated at 37 °C for 30 min before their intramuscular injection. After three hours, a blood sample was collected from the tail of each mouse into heparinized capillary tubes, and the plasma obtained after centrifugation was assayed for creatine kinase (CK) activity using a commercial UV-kinetic kit (CK-NAC Wiener lab) [50].

### 4.7. Inhibition of Hemorrhagic Activity

The minimum hemorrhagic dose was measured for *B. asper* venom. Different venom doses (1, 3, and 5.0 μg) were dissolved in 100 μL of saline solution and injected by the intradermal route into the abdominal skin of four mice. After two hours, animals were euthanized by carbon dioxide inhalation, and their skins were dissected to measure the hemorrhage lesion diameter, according to the protocol described by Kondo et al. [51]. Diameters of hemorrhagic lesions were measured, and the minimum hemorrhagic dose (MHD) was defined as the venom dose that induced a lesion of 10 mm in diameter.

Later, as a positive control, a group of three mice received a dose of two MHD of *B. asper* venom (5.5 µg). For the treatments, separate or mixed compounds were preincubated with two MHD of venom and a dose of 10 mg/kg of inhibitors. As a negative control, two groups were injected with a mix of the inhibitors or saline solution. The hemorrhagic lesions were measured as previously described. For independent injection experiments, groups of four mice were injected intradermally with 5.5 μg of *B. asper* venom, which was preceded by individual or mixed compounds injection at the same site (10 mg/kg) at different time intervals (0, 3, and 5 min).

### 4.8. Molecular Docking Studies

The structure of CP471474 was built using Gauss View 5 [52] (Dennington, Keith, and Millam, 2009). The geometric parameters were optimized with GAUSSIAN 09 [53] using BLYP/3-21G*/DGA1 approximation. Varespladib structure was extracted from PDB 7LYE. Molecular docking was carried out on a personal computer using Autodock Vina [54]. The crystal structures used for this study were the metalloproteinase BaP1 (PDB code 2W15), the PLA_2_ myotoxin I (MT-I) (PDB code 5TFV_A) isolated from *B. asper* venom, and the PLA_2_ from the crotoxin complex (PDB code 2QOG), isolated from *C. d. terrificus* venom. These toxins were selected owing to their relevance in the pathogenesis induced by *B. asper* [55,56] and *Crotalus durissus* subspecies venoms [14,15,57]. Water molecules were removed from the protein, and its structure was prepared using the Protein Preparation module implemented in the Maestro program. Hydrogen atoms were automatically added to each protein according to the chemical nature of each amino acid, based on the ionized form expected in physiological conditions. This module also controls the atomic charges assignment. Each 3D structure of the protein was relaxed through constrained local minimization, using the OPLS force fields to remove possible structural mismatches due to the automatic procedure employed to add the hydrogen atoms. A formal charge of +2 for Zn and Ca ions was assigned, and flexible torsions of ligands were detected. The Zn^2+^ atom was used as the center of the grid (X = 13.589, Y = 16.876, and Z = 23.723). For myotoxin I, the center was the Ca^2+^ cofactor coordinates X = 2.279, Y = 15.924, and Z = 21.732, and for myotoxin II, the coordinates of the nitrogen N1 of His48 imidazole ring (X = 16.056, Y = 1.815, and Z = 17.530) were chosen. The grid size was 24 Å3, and exhaustiveness was 20. Then, the ligand poses with the best affinity were selected, and a visual inspection of the interactions at the active site was performed and recorded. UCSF Chimera (www.cgl.ucsf.edu/chimera/, accessed on 19 September 2022) and Biovia Discovery Studio (https://discover.3ds.com/discovery-studio-visualizer, accessed on 19 September 2022) were used to generate docking images.

### 4.9. Statistical Analysis

With the aim of determining statistical differences in the lethal activity assays, a Chi-square test was performed (alpha was established at 0.05). For the other biological assays, an ANOVA test followed by a Tukey test was applied (compared between all treatments). The statistical analysis was performed with Python (Version 3.10).

## 5. Conclusions

Varespladib and CP471474 partially inhibited the lethal, edema-forming, and myotoxic effects induced by Colombian *B. asper* and *C. d. cumanensis* venoms. In addition, these compounds also reduced the hemorrhage caused by *B. asper* venom. Nevertheless, a synergistic effect was observed when the inhibitors were mixed, but statistical differences were not obtained when the mix was compared with compounds used separately. The combination showed the best inhibitory abilities for independent-injection assays in all the biological experiments. It is the first time that the inhibition of Colombian *B. asper* and *C. d. cumenansis* venoms by varespladib and CP471474 has been shown. In addition, one of the most important results obtained in this study was the partial neutralization of the hemorrhage induced by *B. asper* venom by varespladib, which indicates that this inhibitor also may bind SVMPs, such as is supported by molecular docking results. The results observed in this study may be explained by the molecular modeling results, which proposed that varespladib and CP471474 may bind to the active site of myotoxin I and CB subunit from the crotoxin complex. Moreover, the inhibitors may also occupy the hydrophobic channels of these PLA_2s_. Similar findings were achieved with BaP1 and the inhibitors, which may interact with the active site of this SVMP and occupy the substrate-binding subsites.

## Figures and Tables

**Figure 1 molecules-27-08588-f001:**
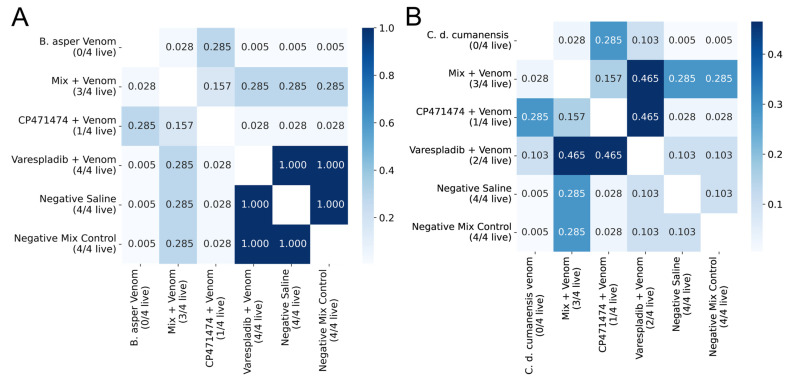
Inhibitory effects of varespladib, CP471474, and their mixture on the lethal activity of Colombian *B. asper* (**A**) and *C. d. cumanensis* (**B**) venoms in preincubation assays. Three DL_50_ were used in the assays, with 10 mg/kg doses for each compound. Values in the squares are *p* values. The darkest blues represent no significant statistical differences (*p* > 0.05), whereas the lightest blue represents statistical differences (*p* < 0.05). *n* = 4.

**Figure 2 molecules-27-08588-f002:**
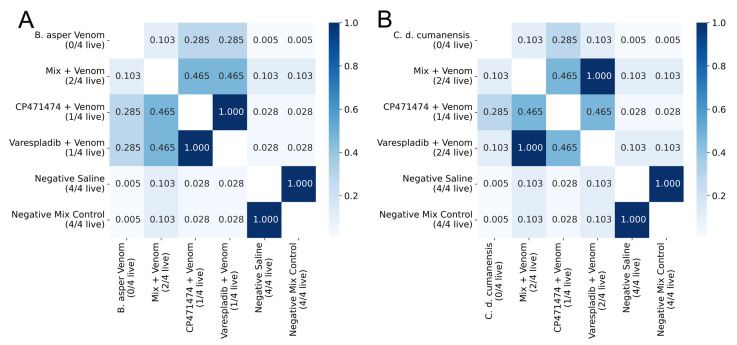
Inhibitory effects of varespladib, CP471474, and their mixture on the lethal activity of Colombian *B. asper* (**A**) and *C. d. cumanensis* (**B**) venoms in independent injection assays. Three DL_50_ of venom were used in the assays and a compound dose of 10 mg/kg. Values in the squares are *p* values. The darkest blues represent no significant statistical differences (*p* > 0.05), whereas the lightest blue represents statistical differences (*p* < 0.05). *n* = 4.

**Figure 3 molecules-27-08588-f003:**
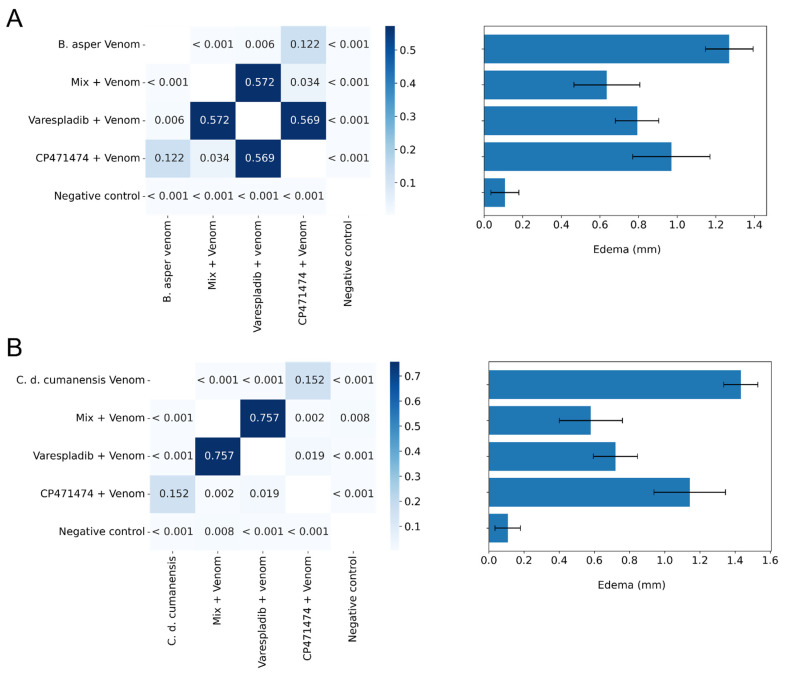
Inhibitory effects of varespladib, CP471474, and their mixture on the edema-forming activity of Colombian *B. asper* (**A**) and *C. d. cumanensis* (**B**) venoms in independent injection assays. A total of 5 μg of the venoms was used in the assays, and 10 mg/kg of a mouse. Values in the squares are *p* values. The darkest blues represent no significant statistical differences (*p* > 0.05), whereas the lightest blue represents statistical differences (*p* < 0.05). Results in the bar graph are presented as mean ± standard deviation. *n* = 4.

**Figure 4 molecules-27-08588-f004:**
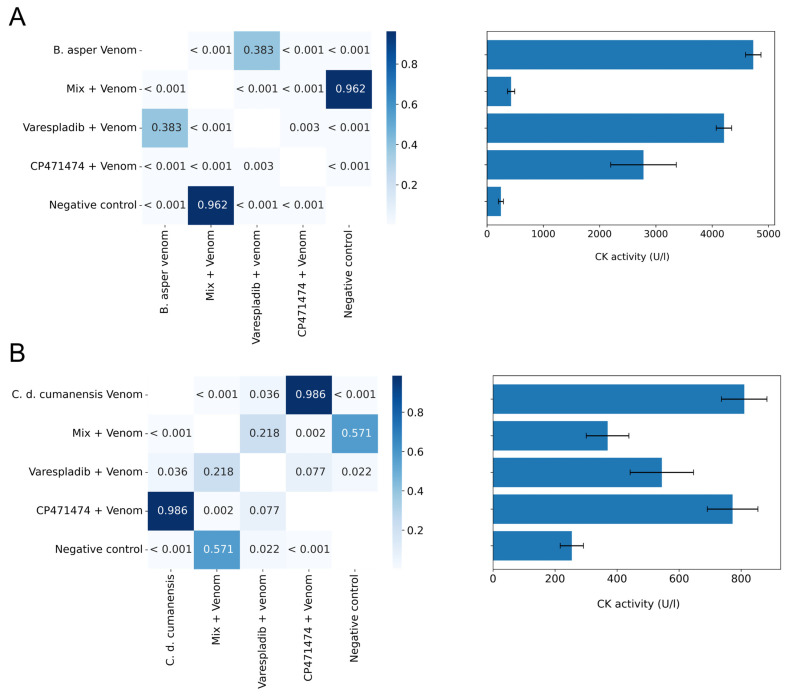
Inhibitory effects of varespladib, CP471474, and their mixture on the myotoxic activity of Colombian *B. asper* (**A**) and *C. d. cumanesis* (**B**) venoms in preincubation assays. Fifty μg of *B. asper* venom or 10 μg of *C. d. cumanensis* venom was used in the assays, and the dose of the inhibitors was 10 mg/kg of mice. Values in the squares are *p* values. The darkest blues represent no significant statistical differences (*p* > 0.05), whereas the lightest blue represents statistical differences (*p* < 0.05). Results in the bar graph are presented as mean ± standard deviation. *n* = 4.

**Figure 5 molecules-27-08588-f005:**
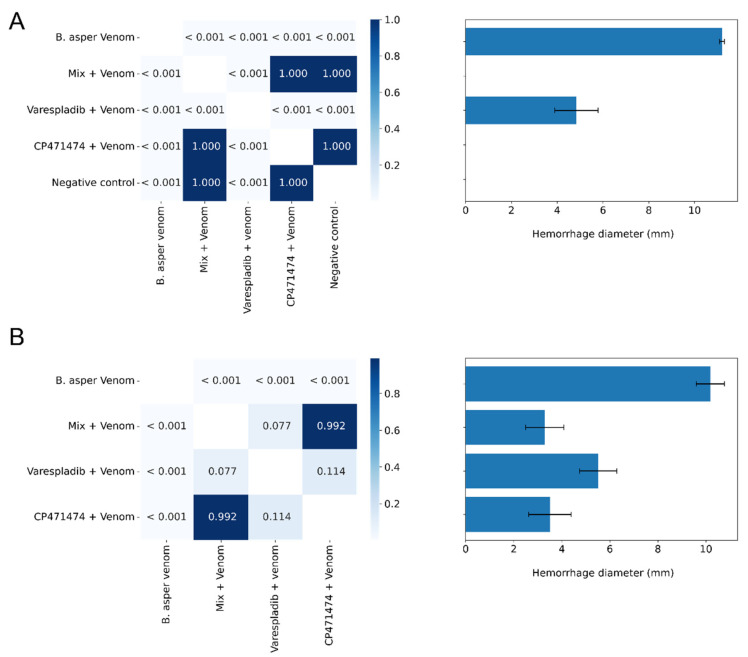
Inhibitory effects of varespladib, CP471474, and a mixture of them on the hemorrhagic activity induced by Colombian *B. asper* venom in preincubation (**A**,**B**) independent injection assays. A total of 5.5 μg of *B. asper* venom (2 MHD) was used in the assays and a dose of 10 mg/kg of each compound. Values in the squares are *p* values. The darkest blues represent no significant statistical differences (*p* > 0.05), whereas the lightest blue represents statistical differences (*p* < 0.05). Results in the bar graph are presented as mean ± standard deviation. *n* = 3.

**Figure 6 molecules-27-08588-f006:**
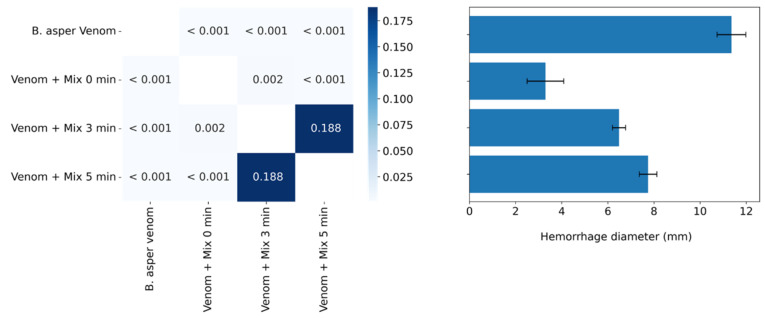
Inhibitory effects of varespladib, CP471474, and their mixture on the hemorrhagic activity induced by Colombian *B. asper* venom in independent injection assays at different time intervals (0, 3, and 5 min). A total of 5.5 μg of *B. asper* venom (2 MHD) was used in the assays, and a dose of 10 mg/kg for each compound. Values in the squares are *p* values. The darkest blues represent no significant statistical differences (*p* > 0.05), whereas the lightest blue represents statistical differences (*p* < 0.05). Results in the bar graph are presented as mean ± standard deviation. *n* = 3.

**Figure 7 molecules-27-08588-f007:**
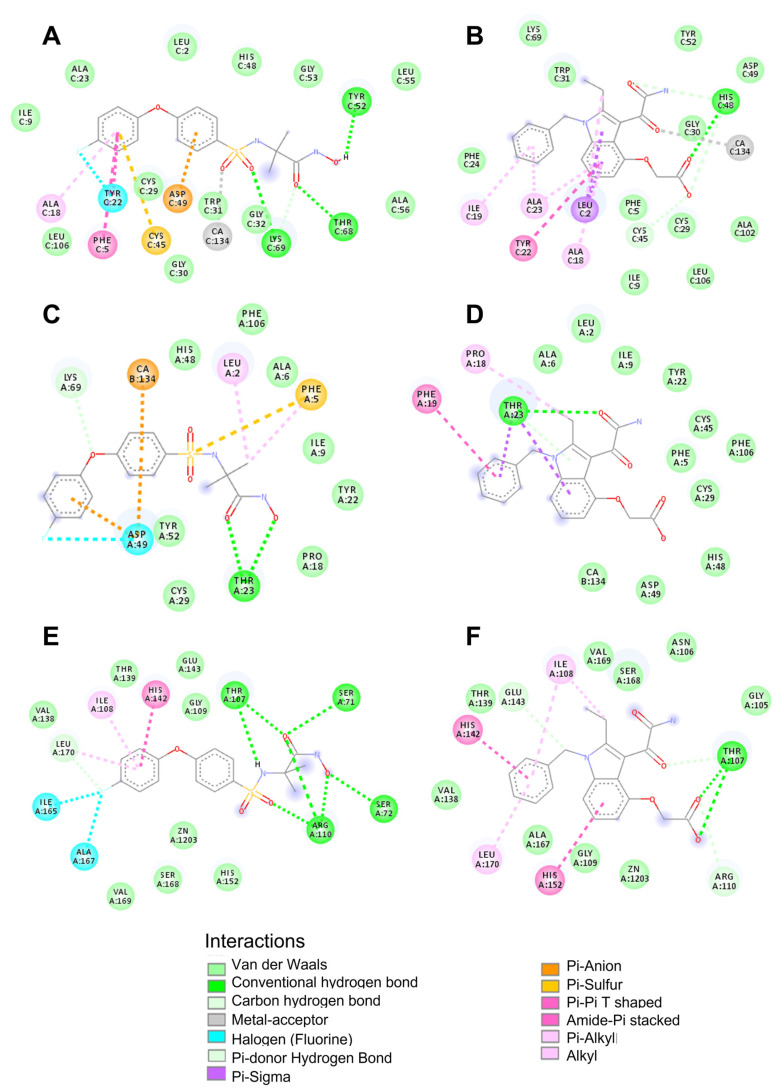
Docking poses with the lowest affinity for PLA_2_ from the crotoxin complex (PDB code 2QOG) in complex with (**A**) CP471474 and (**B**) Varespladib. PLA_2_ myotoxin I (PDB code 5TFV_A) in complex with (**C**) CP471474 and (**D**) Varespladib. Metalloproteinase BaP1 (PDB code 2W15) in complex with (**E**) CP471474 and (**F**) Varespladib. Images were obtained using the available functionalities of Discovery Studio Visualizer.

**Figure 8 molecules-27-08588-f008:**
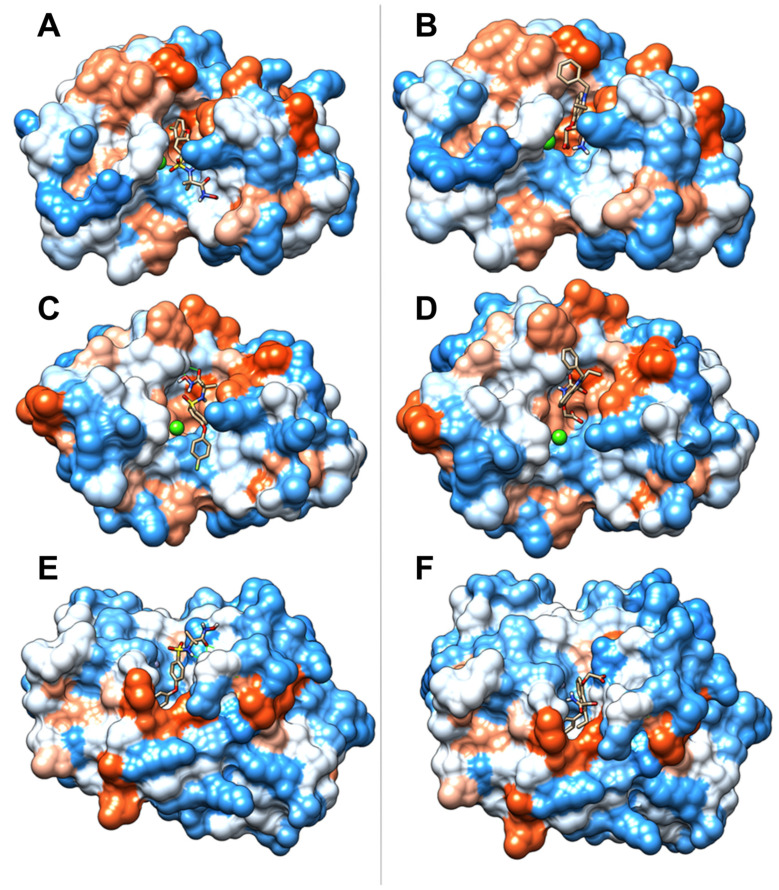
Binding of CP471474 and varespladib to the active site region of studied toxins. PLA_2_ from the crotoxin complex (PDB code 2QOG) in complex with (**A**) CP471474 and (**B**) Varespladib. PLA_2_ myotoxin I (PDB code 5TFV_A) in complex with (**C**) CP471474 and (**D**) Varespladib. Metalloproteinase BaP1 (PDB code 2W15) in complex with (**E**) CP471474 and (**F**) Varespladib. The red areas of the surface represent the acid regions, the white areas represent the neutral, and the blue areas are the basic regions. The green sphere represents Ca^2+^, and the gray sphere is Zn^2+^. Images were obtained using UCSF Chimera.

**Table 1 molecules-27-08588-t001:** Affinities and docking interactions for varespladib and CP471474.

Compound	Protein	Affinity (kcal/mol)	Main Weak Interactions	
			Hidrogen Bonds	Van der Waals
Varespladib	PLA_2_ myotoxin I (PDB code 5TFV_A)	−6.5	Thr23	Ca^2+^, His48,Trp31
	PLA_2_ from the crotoxin complex (PDB code 2QOG)	−8.3	His48	Asp49, Ca^2+^
	Metalloproteinase BaP1 (PDB code 2W15),	−7.6	Thr107	His142, His152
CP471474	PLA_2_ myotoxin I (PDB code 5TFV_A)	−7.1	Thr23	Asp49, Phe5
	PLA_2_ from the crotoxin complex (PDB code 2QOG)	−7.5	Lys49, Thr68, Tyr52	Asp49, Cys45, Ca^2+^
	Metalloproteinase BaP1 (PDB code 2W15)	−8.2	Ser71, Ser72, Thr107, Thr110	His142, Zn^2+^

## Data Availability

Not applicable.

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
