# Peer review of "Inhibitory Effects of Varespladib, CP471474, and Their Potential Synergistic Activity on Bothrops asper and Crotalus durissus cumanensis Venoms"

_molecules, 2022, doi:10.3390/molecules27238588_

Round 1
Reviewer 1 Report
The authors have done a good research and have reasonable results that are of great interest. The paper is interesting and should be published.
Some corrections
1. The authors should improve the abstract to reflect the the major objective of the research. They should also conclude between varespladib and CP471474 and their mixture, which one is most potent. Was there any synergistic effect or antagonistic effect when both were mixed? Also bring out this point in the discussion
2. Add molecular docking to the lkeywords
3. The introduction is good and the authors can bring out what is lacking in the literature that they cited that they wanted to handle. Which problem they wish to solve will make their introduction very nice.
4. Generally, please, review language and avoid very long sentences.
5. Line 239…. 5 µg. Also put the labellings in bold on the figures to make them visible.
6. Enlarge the docking figures to the left.
7. Line 544, confusing… used at the same amounts….what does this mean?
8. Line 546 … µL…. check and correct in other places too.
9. Line 547… death and surviving mice……..
10. Line 570 …. using the same route as for the venom. A third group received a mixture of ….
11. Line 579…. Italic for scientific name
12. Line 530 mg/mL
13. Line 525-526…….Venoms were centrifuged at 800g for 15 min????????
Improve on the conclusion.
Reviewer 2 Report
Summary and recommendation (ID: molecules-2065857)
Envenomation by snakes is one of the major causes of deaths recorded worldwide and have been flagged by the World Health Organization as a neglected tropical disease. In addition, it is a cause of several morbidities and severe socio-economic losses worldwide. Horse-derived antivenin sera is currently the only credited treatment for management of venomous snakebites. Unfortunately, antivenin sera are associated with various side effects, such as development of immediate or delayed hypersensitivity (anaphylaxis or serum sickness) and it does not prevent local tissue damage induced by snake venoms. These side effects are thought to be due to the action of non-immunoglobulin proteins present in high concentrations in the antisera which makes it only effective if administered within 30 minutes to 6 hours of the snakebite. The scourge of snakebites is compounded by poor road networks, fragmented records, lack of public health education, absence of antivenin sera when needed, and poor antivenom preservation facilities in health centres in remote areas where envenomation occurs. For this reason, several attempts have been made to develop snake venom antagonists from other sources including plants, avian eggs, dogs, rabbits, and camelids to supplement those derived from horses.
The current study entitled ‘‘Inhibitory effects of Varespladib, CP471474, and their mixture on Bothrops asper and Crotalus durissus cumanensis venoms from Colombia’’ attempted to evaluate the efficacy of Varespladib (a potent and selective inhibitor of secretory phospholipase A2) and CP471474 (a broad-spectrum peptidomimetic (with a hydroxamate zinc binding group) inhibitor of matrix metalloproteinases) in inhibiting the lethal activities of Bothrops asper and Crotalus durissus cumanensis venoms. The compounds exhibited partial synergistic effects against lethal activity of both venoms, but also individually inhibited the lethal activities of the crude venoms. The authors also performed molecular docking studies to back up their findings.
I should admit that I found the paper to be scientifically written and a significant percentage of it to be well described. I felt confident that the authors performed careful data collection and thorough reporting. However, I found some of the description in the work to be repetitive, while the description of some salient points was inadequate or misplaced. My concerns are detailed below.
2.1. MAJOR COMMENTS
1. Title
I suggest revising the title to: Inhibitory effects of Varespladib and CP471474, and their potential synergistic activity on Bothrops asper and Crotalus durissus cumanensis venoms.
2. Abstract
L17: Antivenoms >> antivenin sera.
L20: ability >> inhibitory effects
L21-22: and a mixture of them >> and their synergistic effects
L23-24: The mix of these compounds partially inhibited the lethal activity of both venoms >> the compounds exhibited partial synergistic effects against lethal activity of both venoms.
-I suggest revising the subsequent lines, by introducing the results of the individual compounds first, and then that of the mixture.
-Molecular docking results need to be included in the abstract.
3. Keywords
Could be revised, avoiding as much as possible those words already used in the title. For increased visibility, I suggest including common names of the snakes (Terciopelo, Venezuelan rattlesnake), and Synergistic antivenoms as author-suggested indexing keywords.
4. Introduction
L43-45: Snake envenomation was added to the list of the neglected tropical diseases by the WHO in March 2009 and later removed (Warrell et al., 2013). With sufficient epidemiological data, the menace was re-included in category A of neglected tropical diseases in June 2017 (Gómez-Betancur et al., 2019).
Warrell et al. (2013). New approaches & technologies of venomics to meet the challenge of human envenoming by snakebites in India. The Indian journal of medical research, 138(1), 38–59.
Gómez-Betancur et al. (2019). Perspective on the Therapeutics of Anti-Snake Venom. Molecules (Basel, Switzerland), 24(18), 3276. https://doi.org/10.3390/molecules24183276
L49: Delete ‘‘sp’’
L104: To bring this case better, you should indicate that the quantity, lethality, and composition of venoms vary with the age and species of the snake, time of the year, geographic location as well as the envenoming snake’s diet. This would suggest the need to study the venoms of these snakes from Colombia.
5. Results
L113: and a mix of them >> and their mixture. Make this correction throughout the manuscript.
Fig. 1 has an error: negaive saline. In the caption, are the authors referring to 3 LD50 as 3 DL50?
Some of these descriptions are not easy to interpret. For example, what do the authors mean by Negative mix?
-Why did the authors decide to plot the p-values of the results, and not the actual values obtained in the various inhibitory assays? The actual results should be presented, and not the p-values.
6. Materials and methods
Sections 4.4 to 4.7 needs to give citation of where these methods were obtained from.
7. Conclusions
L635-637: Both varespladib and CP471474 are already known to be antivenom compounds (see for e.g., Zinenko et al., 2020; Xie et al., 2020 and Preciado et al., 2019). Maybe the authors would wish to indicate that they are partially synergistic in action against the activity of Colombian B. asper and C. d. cumenansis venoms?
Zinenko et al. (2020). PLA2 Inhibitor Varespladib as an Alternative to the Antivenom Treatment for Bites from Nikolsky's Viper Vipera berus nikolskii. Toxins, 12(6), 356. https://doi.org/10.3390/toxins12060356
Xie et al. (2020). Varespladib Inhibits the Phospholipase A2 and Coagulopathic Activities of Venom Components from Hemotoxic Snakes. Biomedicines, 8(6), 165. https://doi.org/10.3390/biomedicines8060165
Preciado et al. (2019). Potential of Matrix Metalloproteinase Inhibitors for the Treatment of Local Tissue Damage Induced by a Type P-I Snake Venom Metalloproteinase. Toxins, 12(1), 8. https://doi.org/10.3390/toxins12010008
Round 2
Reviewer 2 Report
Accept